# Interaction of the Transcription Factors BES1/BZR1 in Plant Growth and Stress Response

**DOI:** 10.3390/ijms25136836

**Published:** 2024-06-21

**Authors:** Xuehua Cao, Yanni Wei, Biaodi Shen, Linchuan Liu, Juan Mao

**Affiliations:** 1State Key Laboratory for Conservation and Utilization of Subtropical Agro-Bioresources, Guangzhou 510642, China; caoxuehua@stu.scau.edu.cn (X.C.); 20222159021@stu.scau.edu.cn (Y.W.); shenbiaodi@stu.scau.edu.cn (B.S.); 2Guangdong Key Laboratory for Innovative Development and Utilization of Forest Plant Germplasm, College of Forestry and Landscape Architecture, South China Agricultural University, Guangzhou 510642, China

**Keywords:** brassinosteroids, transcription factor, Bri1-EMS Suppressor 1, Brassinazole Resistant 1, interaction

## Abstract

Bri1-EMS Suppressor 1 (BES1) and Brassinazole Resistant 1 (BZR1) are two key transcription factors in the brassinosteroid (BR) signaling pathway, serving as crucial integrators that connect various signaling pathways in plants. Extensive genetic and biochemical studies have revealed that BES1 and BZR1, along with other protein factors, form a complex interaction network that governs plant growth, development, and stress tolerance. Among the interactome of BES1 and BZR1, several proteins involved in posttranslational modifications play a key role in modifying the stability, abundance, and transcriptional activity of BES1 and BZR1. This review specifically focuses on the functions and regulatory mechanisms of BES1 and BZR1 protein interactors that are not involved in the posttranslational modifications but are crucial in specific growth and development stages and stress responses. By highlighting the significance of the BZR1 and BES1 interactome, this review sheds light on how it optimizes plant growth, development, and stress responses.

## 1. Introduction

Brassinosteroids (BRs) are plant-specific polyhydroxylated steroidal hormones that play a crucial role in regulating various aspects of plant growth and development events, as well as in responses to pathogens and environmental stresses. Extensive studies in Arabidopsis (*Arabidopsis thaliana*) over the past two and a half decades have significantly enhanced our understanding of the BR signaling pathway. In Arabidopsis, the BR receptor BRI1 (Brassinosteroid Insensitive 1) [1,2,3], along with its homologs BRL1 (BRI1-Like 1) and BRL3 [4,5], and its coreceptor BAK1 (BRI1-Associated Receptor Kinase), perceive BR ligands and facilitate their transphosphorylation and activation [6,7,8,9,10]. The binding of BR also leads to the release of BKI1 (BRI1 Kinase Inhibitor 1), a negative regulator of BR signaling, from BRI1 [11]. The activated BRI1-BAK1 complex initiates a cascade of phosphorylation and dephosphorylation events, ultimately leading to the dephosphorylation and inactivation of the GSK3-like kinase BIN2 (Brassinosteroid-Insensitive2) [12]. The inactivated BIN2 cannot phosphorylate the master positive transcription factors BES1 (Bri1-EMS Suppressor1, also known as BZR2) and BZR1 (Brassinazole Resistant 1) in the BR signaling pathway [13,14]. Once dephosphorylated by PP2A (Protein Phosphatase 2A) [15], the active BES1 and BZR1 directly or indirectly regulate the transcription of target genes by binding to E-box (CANNTG) elements and the BRRE (CGTGT/CG) motif in their promoter region, eventually influencing plant growth and development as well as stress responses [16,17,18,19,20,21]. In the absence of BR, BIN2 is activated and phosphorylates BES1 and BZR1, leading to their sequestration in the cytoplasm by 14-3-3 proteins [22]. As a result, their nuclear localization and DNA-binding activity are hindered, and the degradation of BES1/BZR1 is facilitated [7,23].

As key transcription factors, BES1 and BZR1 undergo various posttranslational modifications that affect their activity and protein levels, playing a crucial role in monitoring the output of BR signaling. For instance, PP2A and BIN2, well-established regulators of the BR signaling pathway, directly interact with BES1/BZR1 to modulate their dephosphorylation and phosphorylation status. Dephosphorylation by PP2A activates BES1/BZR1, leading to the upregulation of BR-responsive genes. Conversely, phosphorylation of BES1/BZR1 by BIN2 triggers their degradation with the assistance of 14-3-3 protein, thereby negatively influencing BR signaling as well as plant growth and development [14,15,22,24]. Additionally, several E3 ligases, such as SINAT (Seven-IN-Absentia of *Arabidopsis thaliana*), COP1 (Constitutive Photomorphogenic 1), MAX2 (More Axillary Growth Locus 2), and PUB40 (Plant U-box 40) in Arabidopsis, along with OsPUB24 in rice (*Oryza sativa*), interact with BES1/BZR1 to facilitate their degradation [25,26,27,28,29]. Moreover, BZR1 and BES1 also have the ability to integrate internal and external signals by interacting with key components of other signaling pathways, thereby co-regulating plant growth and stress tolerance [30,31,32] (Figure 1).

Previous reviews have mainly focused on the BR signal transduction and its regulatory mechanism, rather than specifically on the interactome of BES1/BZR1. Understanding this interactome is crucial for elucidating the intricate regulatory network of plant growth and development under both normal and diverse growth conditions and can also offer valuable insights for crop improvement through gene editing or molecular breeding. This review specifically highlights the roles of the BES1/BZR1 interactome in optimizing plant growth, development, and stress tolerance, while excluding components of the BR signaling pathway and posttranslational modifiers.

## 2. Interactors of BES1/BZR1 in Plant Growth and Development

BES1 and BZR1 have the ability to form a variety of protein complexes with other transcription factors to collaboratively regulate plant growth and stress responses. For example, BES1 physically interacts with bHLH family protein BIMs (BES1 Interacting MYC-Like proteins), MYB (Myeloblastosis) family proteins MYB30 and MYBL2 (Myeloblastosis Family Transcription Factor-Like 2), as well as HDZIP (homeodomain-leucine zipper) transcription factor HAT1 (Homeobox *Arabidopsis Thaliana* 1) [21,33,34,35]. These interactions can either enhance or inhibit the expression of BR-responsive genes, thereby coordinating plant growth and development. Moreover, BES1/BZR1 also interacts with key protein factors involved in light and other plant hormone signaling pathways, allowing them to integrate diverse signals and control specific plant developmental processes (Table 1).

### 2.1. Interactors of BES1/BZR1 in Skotomorphogenesis and Photomorphogenesis

In darkness, plants undergo skotomorphogenesis, during which most dicot seedlings develop an etiolated apical structure consisting of an apical hook, unexpanded cotyledons, and a greatly elongated hypocotyl due to cell expansion [61]. The BAP regulatory module, consisting of the BR-signaling transcription factor BZR1, the auxin-response factor ARF6 (Auxin Response Factor 6), and the light/temperature-regulated transcription factor PIF4 (Phytochrome-Interacting Factor 4), synergistically interacts to control cell elongation and hypocotyl growth by regulating common target genes involved in cell wall modifications and auxin responses [36]. Therefore, BR integrates various signals, including auxin, light, and temperature, through BZR1 and its interacting partners to play a crucial role in hypocotyl development in Arabidopsis. In addition to the typical BAP module, other activators and suppressors, such as BIC1 (Blue Light Inhibitor of Cryptochromes 1), also contribute to regulating hypocotyl elongation by interacting with BES1/BZR1. BIC1 acts as a transcriptional coactivator for BZR1 and PIF4, potentially increasing their access to target gene promoters and synergistically activating genes related to cell elongation, thus stimulating both cell and hypocotyl elongation [37]. Interestingly, the interaction between BIC1 and BZR1 is also evolutionarily conserved in the model monocot plant *Triticum aestivum* (bread wheat), shedding light on the molecular mechanisms understanding the coordination between BR and light signaling pathway across different plant species [37].

BR and GA mutually regulate each other to modulate skotomorphogenesis. GA is known to play a role as a positive regulator in skotomorphogenesis. The plant-specific protein BLI (BLISTER) probably acts as a scaffold and interacts with BZR1 through its C-terminal coiled-coil domain to enhance the expression of BZR1-dependent BR-responsive genes, promoting hypocotyl elongation in darkness [38]. Moreover, BLI and BZR1 have been found to co-regulate skotomorphogenesis by orchestrating the expression of gibberellin (GA) biosynthetic genes, ultimately leading to the production of bioactive GAs. These findings emphasize the collaborative role of BLI and BZR1 in regulating skotomorphogenesis by enhancing BR signaling and GA biosynthesis. In Arabidopsis, DELLAs are known as negative regulators of GA signaling and also act as negative regulators of BZR1. GA can trigger the degradation of DELLAs [39,62]. DELLAs counteract BZR1 through direct interaction, thereby attenuating BZR1’s transcriptional activity and protein stability. As a result, the expression of growth-related genes and cell elongation are repressed [39,62]. HSP90 (Heat Shock Protein 90) interacts physically with DELLA and BZR1 to regulate the degradation and abundance of DELLA and control the expression of BZR1 target genes, ultimately facilitating hypocotyl elongation [40]. The degradation of DELLA is dependent on the ATPase activity of HSP90, which is induced by GA. On the other hand, inhibiting HSP90 bioactivity leads to the stabilization of the DELLA/BZR1 complex, eventually affecting Arabidopsis hypocotyl growth through gene expression regulation. In a similar manner, in tomato *(Solanum lycopersicum*), there is a reciprocal regulation between SlBES1.8 and GA to influence tomato growth and development [63]. SlBES1.8 mediates GA deactivation and signaling by directly regulating the expression of genes involved in GA deactivation and receptor genes, impacting leaf morphogenesis and SAM (shoot apical meristem) development. Furthermore, SlDELLA inhibits the transcriptional regulatory activity of SlBES1.8 by directly interacting with SlBES1.8, thus preventing DNA binding of SlBES1.8 [63].

During skotomorphogenesis, apical hook development is regulated by BES1/BZR1 in conjunction with various components of light and other plant hormone signaling pathways, such as ET (ethylene), auxin, and JA (jasmonates). BZR1 directly interacts with EIN3 (Ethylene-Insensitive 3), a positive transcription factor in ethylene signaling, to promote the essential regulator *HLS1* (*Hookless1*) for hook development. This indicates a cooperative role of BR and ET in promoting apical hook development [41]. Moreover, *SAUR17* (*Small Auxin Up RNA17*), which is rapidly responsive to auxin treatment, acts as a positive regulator for apical hook and closed cotyledon formation in an apical organ-specific manner. In darkness, *SAUR17* expression is stimulated by PIFs-EIN3-EIL1-BZR1 transcription factor complex. Within this complex, EIN3 and PIFs play crucial roles in enhancing the DNA-binding ability of BZR1, leading to the activation of *SAUR17* transcription [42]. JA-dependent transcription factors MYC2, MYC3, and MYC4 regulate BR biosynthesis by repressing the BR biosynthetic gene *DWF4* (*DWARF4*). Additionally, they can interfere with the association between BZR1 and PIFs through the MYC2-BZR1 interaction. This disruption ultimately leads to the downregulation of *WAG2* (*Wavy Root Growth 2*), a key kinase in apical hook formation, thus negatively impacting apical hook development [43].

The transition from skotomorphogenesis to photomorphogenesis, a vital process for seedling survival upon emergence from the soil, involves the collaboration of BZR1, PIF4, and their interactor GRF7 (Growth-Regulating Factor 7). They collaborate to repress the expression of critical enzymes involved in chlorophyll biosynthesis to prevent photo-oxidation damage while also promoting the expression of genes responsible for cell elongation, ultimately enhancing seedling survival upon light exposure. Furthermore, the expression of *GRF7* is induced by BZR1 and PIF4 [44].

When seedlings grow through the soil, they undergo photomorphogenesis in the presence of light. Increasing evidence has revealed that BES1/BZR1 interacts with various protein factors to co-regulate photomorphogenesis. The photoreceptors phyB (Phytochrome B, red light receptor), CRY1 (Cryptochrome 1, blue light receptor), and UVR8 (UV Resistance Locus 8, UV-B light photoreceptor) have been demonstrated to interact with BES1 and/or BZR1, repressing their DNA-binding activity and thereby regulating photomorphogenesis mediated by BR [45,46,47,48]. Moreover, CRY1 physically interacts with the BES1 interactor BIM1 (BES1-Interacting MYC-Like 1), which in turn promotes the activity of BES1. Specifically, under blue light, CRY1 directly interacts with the dephosphorylated and activated BES1, ultimately disrupting its DNA-binding activity and the expression of its target genes [46]. CRY1 also negatively regulates the function of BZR1 through two mechanisms. Firstly, it interacts with BZR1 on the DNA-binding domain, suppressing its DNA-binding and transcriptional activity. Additionally, CRY1 interacts with BIN2 and facilitates the association of BIN2 and BZR1 in a light-dependent manner, resulting in increased phosphorylation of BZR1 and the repression of BR-mediated plant growth and development. These findings shed light on the intricate coordination of plant growth and development in Arabidopsis through the interplay between blue light and BR signaling [47]. UVR8 physically interacts with dephosphorylated BES1 to inhibit its DNA-binding activity, thereby regulating the expression of growth-related genes controlled by BES1 [48]. These findings emphasize the significance of the UVR8-BES1 interaction as a critical module that integrates light and BR signaling. phyB directly interacts with the DNA-binding domain of BZR1, affecting its function and repressing BR signaling [45]. In addition to key transcription factors in light signaling, BBX20/BZS1 (B-Box Domain Protein 20) and BBX32, both B-box-containing transcription factors, also contribute to BR and light-regulated photomorphogenesis [49,64,65]. Initially identified as a negative regulator of photomorphogenesis due to its role in promoting hypocotyl elongation [65], BBX32 interacts with BZR1 and PIF3 to regulate the expression of shared target genes that control cotyledon opening and closing [49]. This discovery enhances our understanding of the regulatory mechanism involved in cotyledon opening in the process of de-etiolation.

As shown above, the interaction between BES1 and PIF4, along with their regulatory functions, has been extensively studied in the context of BES1/BZR1 interactors. PIF4 and BZR1 play crucial roles in integrating light and BR signaling pathways, with their interaction being modulated by various protein factors. This interaction is crucial for regulating different plant developmental processes, including hypocotyl elongation, apical hook development, and de-etiolation. Additionally, higher temperatures trigger the translocation of BZR1 into the nucleus and boost *PIF4* transcription, leading to the co-activation of growth-promoting genes [66]. These results highlight the collaborative role of PIF4 and BZR1 in coordinating plant growth in both normal and challenging environments. Nevertheless, additional research is necessary to fully elucidate the molecular regulatory mechanisms involving BES1/BZR1 and their interacting partners. This will provide valuable insights into the synergistic regulation of plant growth by BR and other signaling pathways. Such investigations will contribute to unraveling the complex regulatory network that governs plant growth and development in varying light conditions as well as in response to stress.

### 2.2. Interactors of BES1/BZR1 in Root Growth

In Arabidopsis, the root meristem cells go through a series of processes, including cell division, cell expansion, and terminal cell differentiation, to produce different functional cell types. These processes rely on precise spatiotemporal instructions. Previous studies have extensively investigated various root developmental transitions and their regulatory mechanisms in Arabidopsis. BR, in antagonism with other plant hormones, plays a crucial role in maintaining the balance of root stem cells and in determining the transition from cell division in the meristem to cell elongation [67]. *BZR1* exhibits a spatially specific expression pattern, showing low levels in stem cells and high levels in the transition-elongation zone in Arabidopsis roots [67]. As a result, BZR1 mainly promotes the expression of its target genes in the transition-elongation zone while repressing genes in the vicinity of stem cells. On the other hand, auxin has an opposing effect on gene expression in these specific regions [67]. Hence, BR and auxin counteract each other in these areas, resulting in varied cellular responses and behaviors.

In addition to the co-regulation of root development by plant hormones, BES1/BZR1 have been found to interact and collaborate with SHR (Short-Root) and BRAVO (Brassinosteroids at Vascular and Organizing Center), two well-known master transcription factors in root growth and development. BZR1 is highly enriched in the elongation zone but decreases in the mature zone where lignification starts, indicating a potential negative regulation in initiating lignified cells. The spatial distribution of nuclear BZR1 aligns with the specific function of BR signaling, influencing developmental patterns by regulating gene expression related to cell expansion and differentiation, as well as modulating SHR activity through direct interaction with SHR [50,68]. The high levels of BZR1 in the transition-elongation zone promote cell expansion-related gene expression while repressing the expression of genes related to cell differentiation. Additionally, BZR1 physically interacts with SHR, inhibiting its activity and suppressing the expression of SHR target genes *CASP1* and *VND7*, which are crucial for lignification in the endodermis and protoxylem [50]. On the other hand, the rapid decrease in BZR1 levels in the mature zone of the root leads to increased expression of BR-suppressed genes related to cell differentiation, including root lignification, and releases inhibition on SHR [50]. This process ultimately leads to the formation of root lignification in the specific region. During the development of root ground tissue, the BZR1 and SHR transcriptional complex enhances the expression of downstream genes, such as the cell cycle gene *CYCD6;1* (*CYCLIND6;1*), which promotes periclinal division [51,68]. Furthermore, the interaction between BZR1 and SHR integrates BR and redox signals to further stimulate periclinal division during root ground tissue maturation by upregulating *RBOHs* (*Respiratory Burst Oxidase Homologs*). This leads to elevated H_2_O_2_ levels, which in turn strengthens the interaction between BZR1 and SHR and contributes to the development of ground tissue [51].

It has been shown that the R2R3-MYB transcription factor, BRAVO, negatively impacts the divisions of the QC (quiescent center) in the primary root of Arabidopsis. Additionally, it has been discovered that BES1 physically interacts with BRAVO and suppresses its transcriptional activity, shedding light on how BR signaling affects the quiescence of plant stem cells [52]. These findings imply that BES1/BZR1, in conjunction with other protein factors, controls root development in a specific tissue-dependent manner and that the strength of BR signaling influences certain biological processes.

### 2.3. Interactors of BES1/BZR1 in Other Developmental Processes

One of the most distinguishing characteristics of BR is its role in promoting cell elongation. This biological process is mainly controlled by the direct regulation of genes related to cell elongation by BZR1/BES1. The G-protein β subunit AGB1 and the chromatin-remodeling factor PKL (PICKLE) interact with BZR1 to facilitate the expression of genes involved in cell elongation [53,54]. Specifically, the CCCH zinc finger protein C3H15 interacts with BES1/BZR1 and they antagonize each other to control the expression of their shared target gene, *SAUR15* (*Small Auxin-Up RNA 15*), which is involved in cell elongation [55]. These findings indicate an antagonistic regulation in BR signaling for the control of cell elongation, highlighting our understanding of the regulatory processes in plant cell development induced by BR.

Multiple lines of evidence have emphasized the significant roles of BES1/BZR1 and their associated proteins in various aspects of plant reproduction, such as flowering, ovule development, pollen production, fruit ripening, and seed formation. Both CK (cytokinin) and BR have been demonstrated to increase seed numbers and contribute to seed yield. It has been observed that the CK-activated transcription factor ARR1 (Arabidopsis Response Regulator 1) directly interacts with BZR1, facilitating ovule initiation and enhancing seed production by regulating ARR1-targeted gene expression. This discovery provides insight into how BR enhances seed yield by synergizing with and boosting CK activity [69]. In Arabidopsis, the interaction between the cyclophilin protein CYP20-2 and BZR1 not only enhances the phosphorylation and degradation of BZR1 but also alters its secondary structure. This alteration leads to the release of transcriptional repression on *FLD* (*Flowering Locus D*), ultimately affecting flowering [56]. In pear (*Pyrus ussuriensis*), BR inhibits ethylene biosynthesis during fruit ripening by suppressing the activity of ACO1 (ACC oxidase1), a key enzyme in ethylene biosynthesis, through interaction with BZR1 [57]. Additionally, in rice, OsWRKY53 positively regulates BR signaling by interacting with BZR1, resulting in the coordinated regulation of rice architecture and seed size development [58,59]. The mediator complex, composed of multiple subunits, acts as a link between specific transcription factors and RNA polymerase II (Pol II), thereby facilitating the transcription of Pol II-regulated genes. In rice, a mediator complex subunit called OsMED25 functions as a co-suppressor of OsBZR1, influencing the expression of numerous OsBZR1-regulated genes and consequently impacting the architecture of rice plants [60]. These findings contribute to our understanding of the role of BES1/BZR1 and its related mechanisms during the vegetative stage of plant growth, providing valuable insights for improving crop yield.

## 3. Interactors of BES1/BZR1 in Stress Response

### 3.1. Interactors of BES1/BZR1 in Abiotic Stress Response

In addition to their roles in plant growth and development, BES1/BZR1 also interact with other protein factors to regulate plant stress responses. Drought, a common abiotic stress, can have detrimental effects on plant growth and survival. Under normal conditions, the WRKY family transcription factors WRKY46, WRKY54, and WRKY70 interact with BES1 to promote the expression of genes related to plant growth and inhibit genes influenced by dehydration, ultimately supporting plant growth and dampening the drought response [70]. However, under drought stress conditions, the degradation of BES1 and WRKY54 is accelerated, thus reducing their impact on plant growth and resulting in increased drought responses [70]. Additionally, in drought conditions, the NAC (No apical meristem, Arabidopsis transcription activation factor, and Cup-shaped cotyledon) family transcription factor RD26 (Responsive to Desiccation 26) and the AP2/ERF transcription factor TINY directly interact with BES1. These interactions allow RD26 and TINY to repress the transcriptional activity of BES1 in regulating BR-induced genes related to plant growth [71,72]. Simultaneously, both TINY and RD26 facilitate the expression of drought-responsive genes, thereby improving plant survival during drought stress [72,73]. These findings provide insights into a mechanism that enhances plant growth during drought stress. It is worth noting that the expression of *RD26* and *TINY* can also be induced by other stresses, like salt. However, the impacts of the interplay between TINY/RD26 and BES1/BZR1 on maintaining a balance between salt tolerance and plant growth are still not fully understood.

Heat stress induces the expression of *HSPs* in plants, improving their resilience in challenging growth conditions. When exposed to heat stress, BES1 is activated and contributes to enhancing the heat response by directly binding to HSEs (Heat Shock Elements) found in HSFs (Heat Shock Transcription Factors), such as HSFA1. This interaction ultimately leads to the activation of genes regulated by heat shock, offering defense against heat stress [74].

Nitrogen (N) is an essential nutrient for plant growth and crop yields. When nitrogen availability is limited, it triggers changes in root morphology and architecture. A decrease in nitrate concentration significantly boosts *BES1* transcript levels, particularly in roots, leading to high levels of dephosphorylated BES1. The activated BES1 then triggers the expression of key genes essential for nitrate uptake and assimilation, thereby stimulating the root’s response to limited nitrogen levels. Additionally, BES1 interacts with LBD37 (Lateral Organ Boundary Domain 37), which is known as a negative regulator of nitrogen signaling [75]. This interaction inhibits the DNA-binding activity of LBD37, leading to the promotion of the expression of several nitrogen-responsive genes that play a role in regulating the nitrogen response. These findings shed light on a new mechanism by which BR signaling influences the root system’s response to low nitrogen levels [75,76]. NH_4_^+^ is the primary form of nitrogen absorbed by higher plants from the soil, playing a crucial role in rice yield. The uptake of NH_4_^+^ into roots is facilitated by AMT-type NH_4_^+^ transporters, specifically OsAMT1;1 (Ammonium Transporter 1;1), OsAMT1;2, and OsAMT1;3, which are of particular importance [77,78]. In rice, the transcription factor IDD10 (Indeterminate Domain 10) directly activates the expression of *OsAMT1;2* [79]. In addition, OsBZR1 regulates the expression of *OsAMT1;2*, either independently or through interaction with IDD10, enhancing NH_4_^+^ uptake in rice roots [80,81]. Furthermore, phyB interacts with both IDD10 and BZR1, inhibiting their DNA-binding activity and the expression of *AMT1;2* regulated by IDD10-BZR1. Ultimately, these interactions modulate NH_4_^+^ uptake, which is crucial for crop yield improvement and rice resistance against sheath blight (ShB) and saline-alkaline stress [81].

### 3.2. Interactors of BES1/BZR1 in Biotic Stress Response

Several factors have been reported to interact with BES1/BZR1 in regulating plant biotic stress responses. For example, when exposed to insect (herbivory *S. exigua*), BES1 physically interacts with MYB family transcription factors MYB34, MYB51, and MYB122 in the nucleus to suppress the biosynthesis of JA-induced insect defense-related metabolites. This indicates that BR counteracts JA-activated plant defense responses [82]. In addition to its role in regulating stress responses through interactions with transcription factors in the nucleus, BZR1, which is localized in both the nucleus and cytoplasm, also plays a crucial role in plant immunity by interacting with other protein factors in the cytoplasm. EDS1 (Enhanced Disease Susceptibility 1), serving as an essential positive regulator of plant innate immunity, interacts with both cytoplasmic and nuclear BZR1 [83]. The interaction between cytoplasmic BZR1 and EDS1 facilitates the dissociation of EDS1 and RPS4, leading to the activation of RPS4-controlled ETI (effector-triggered immunity). This discovery sheds light on a previously undiscovered role for cytoplasm-localized BZR1. Additionally, the interaction between nuclear BZR1 and EDS1 suppresses the transcriptional activity of BZR1, leading to the inhibition of BZR1-activated genes and BR-promoted growth [83]. These findings illustrate the role of BZR1 in balancing plant growth and defense. Notably, the cytoplasmic accumulation of mutant BZR1 enhances pathogen resistance without affecting plant growth, suggesting that the BZR1-EDS1 module could potentially improve both crop productivity and pathogen resistance simultaneously. Furthermore, the interaction between phyB and BZR1 extends beyond plant photomorphogenesis to influence plant responses to biotic stress. In rice, the NAC transcription factor NAC028 collaborates with BZR1 to activate *CAD8B* expression, which is a crucial enzyme in the lignin biosynthesis pathway that contributes to resistance against ShB. Conversely, phyB interacts with BZR1 to suppress the BZR1-NAC028-CAD8B signaling pathway, leading to reduced resistance of rice to ShB [84].

In cotton (*Gossypium hirsutum*), the *GhTINY2* gene encodes the homologous protein of AtTINY in Arabidopsis, which is known to play a role in drought stress. GhTINY2 functions by promoting the accumulation of SA (salicylic acid) and SA signaling transduction, crucial for plant defense responses. Additionally, GhTINY2 interacts with BZR1, negatively regulating its transcriptional activity and BR signaling, resulting in reduced plant growth but increased immune response [85]. These discoveries, coupled with the function of AtTINY-BES1 in drought stress in Arabidopsis, reveal a tradeoff mechanism between plant growth and responses to biotic and abiotic stress mediated by TINY-BES1/BZR1.

Previous studies have already confirmed the significant roles of BES1/BZR1 in responding to a variety of stresses. However, further investigations are imperative to gain a comprehensive understanding of how BES1/BZR1 is regulated by various interactors in different environmental stress conditions. Additionally, uncovering more interactors of BES1/BZR1 and their regulatory networks in diverse environmental settings can provide valuable insights into the regulation of plant growth and stress response. The interactors of BES1/BZR1 involved in the regulation of plant stress responses are listed in Table 2.

## 4. Conclusions

BES1/BZR1 proteins act as central hubs connecting various signaling pathways to regulate plant growth, development, and stress responses. Experimental evidence has established multiple physical connections between BES1/BZR1 and their interacting partners involved in a range of developmental processes and stress responses. However, the dynamic and spatiotemporal organization of these molecular interactions remain poorly understood. A variety of interactors from different signaling pathways have been reported to coordinate with BES1/BZR1 in a tissue-specific and developmental stage-dependent manner to control plant growth, raising fundamental questions about how BES1/BZR1 determine their interacting partners and how these proteins establish a complex regulatory network to carry out their functions. It is also crucial to investigate whether additional regulators from other pathways interact with BES1/BZR1 to modulate more cellular or developmental processes responsive to BR. BR, a growth-promoting hormone, regulates important agronomic traits in crops, such as plant height, seed size, and stress tolerance. Manipulating the BR response pathway represents a promising approach to boost crop yield and stress tolerance [86]. While our current knowledge of the interplay of BES1/BZR1 is largely based on studies in the model plant Arabidopsis, further exploration of these interactions in crops is essential to uncover potential applications for improving crop productivity and enhancing stress tolerance. Moreover, research has shown that BES1/BZR1 family transcription factors are directly relevant to several important agronomic traits in a few plant species. Future studies should explore their potential for crop improvement through gene editing or molecular breeding.

## Figures and Tables

**Figure 1 ijms-25-06836-f001:**
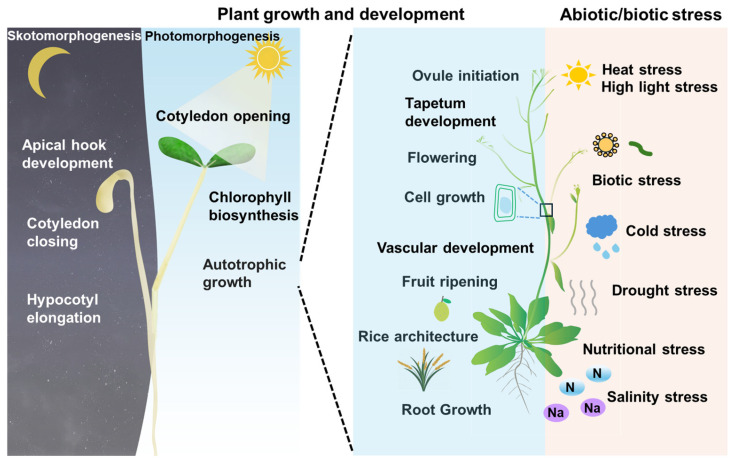
BES1/BZR1 play a crucial role in regulating various aspects of plant growth, development, and stress resistance. They exert a wide range of effects on different plant parts, including roots, stems, leaves, and flowers, throughout the plant’s life cycle. Initially, they are involved in skotomorphogenesis after seed germination in dark conditions (left panel colored with gray), transitioning to photomorphogenesis as seedlings emerge into the light (left panel colored with blue). Subsequently, they contribute to all stages of plant growth (right panel colored with blue) and play a role in enhancing tolerance to multiple stresses (right panel colored with pink).

**Table 1 ijms-25-06836-t001:** Involvement of the interaction proteins of BES1/BZR1 in the regulation of plant growth and development.

Gene Name	Locus	Effect on Plant Growth through Interaction with BES1/BZR1	References
*ARF6*	AT1G30330	Promoting cell elongation and hypocotyl growth	[36]
*PIF4*	AT2G43010	Promoting cell elongation and hypocotyl growth	[36]
*BIC1*	AT3G52740	Promoting cell and hypocotyl elongation	[37]
*BLI*	AT3G23980	Promoting hypocotyl elongation in darkness	[38]
*RGA*	AT2G01570	Inhibiting cell elongation and BR-regulated plant growth	[39]
*HSP90*	AT5G52640	Promoting hypocotyl elongation	[40]
*EIN3*	AT3G20770	Promoting apical hook development	[41]
*SAUR17*	AT4G09530	Promoting apical hook and closed cotyledon	[42]
*WAG2*	AT3G14370	Inhibiting apical hook development	[43]
*GRF7*	AT5G53660	Repressing chlorophyll biosynthesis promoting cell elongation	[44]
*phyB*	AT2G18790	Repressing BR signaling	[45]
*CRY1*	AT4G08920	Inhibiting hypocotyl elongation under blue light	[46,47]
*UVR8*	AT5G63860	Repressing BR-regulated photomorphogenesis	[48]
*BBX32*	AT3G21150	Inhibiting cotyledon opening	[49]
*SHR*	AT4G37650	Suppressing root lignification promoting periclinal division	[50,51]
*BRAVO*	AT5G17800	Suppressing root quiescent center division	[52]
*AGB1*	AT4G34460	Promoting cell elongation	[53]
*PKL*	AT2G25170	Promoting cell elongation	[54]
*C3H15*	AT1G68200	Inhibiting cell elongation	[55]
*CYP20-2*	AT5G13120	Promoting flowering	[56]
*ACO1*	AT2G19590	Promoting fruit ripening	[57]
*OsWRKY53*	Os05g0343400	Regulating rice architecture and increasing seed size	[58,59]
*OsMED25*	Os09g0306700	Regulating rice architecture	[60]

**Table 2 ijms-25-06836-t002:** Involvement of the interaction proteins of BES1/BZR1 in the regulation of stress responses.

Gene Name	Locus	Effect on Stress Tolerance through Interaction with BES1/BZR1	References
*WRKY46*	AT2G46400	Suppressing drought response	[70]
*WRKY54*	AT2G40750	Suppressing drought response	
*WRKY70*	AT3G56400	Suppressing drought response	
*RD26*	AT4G27410	Inhibiting BR-regulated growth under drought condition	[71]
*TINY*	AT5G25810	Inhibiting BR-regulated growth under drought condition	[72]
*HSFA1a*	AT4G17750	Improving heat stress	[74]
*LBD37*	AT5G67420	Promoting nitrogen response	[76]
*OsIDD10*	Os04g47860	Improving nitrogen uptake	[81]
*phyB*	AT2G18790	Inhibiting nitrogen uptake and sheath blight resistance	[81,84]
*MYB34*	AT5G60890	Suppressing insect defense	[82]
*MYB122*	AT1G74080	Suppressing insect defense	[82]
*MYB51*	AT1G18570	Suppressing insect defense	[82]
*EDS1*	AT3G48090	Increasing pathogen resistance	[83]
*GhTINY2*	GhD06G0642	Increased immune response	[85]

## Data Availability

No new data were created or analyzed in this study.

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
