# Peer review of "Interaction of the Transcription Factors BES1/BZR1 in Plant Growth and Stress Response"

_ijms, 2024, doi:10.3390/ijms25136836_

Round 1

Reviewer 1 Report

Comments and Suggestions for Authors

The paper is well written and the information is complete and comprehensible. The major problem is that is an only text review, and this decreases the interest. Most readers are looking for figures or tables that facilitate the understanding of the question. In the present version there is only a single table, with a very bad selection of colors, and not really informative. I recommend to complete this review as follows:

a) Include one table summarizing all the published information of BES/BZR1 on plant growth and development, including the effect, the plant where was observed, and the proper citation. 

b) Include one table summarizing all the published information of BES/BZR1 on biotic and abiotic stress, including the effect, the plant where was observed, and the proper citation. 

c) Include a figure with one plant, indicating in which parts of the plant anatomy the BES/BZR1 have some effect.

Minor points: line 328. "S. exigua" should go in italics and with spacing.

Author Response

Dear Editor and Reviewers,

Thank you very much for handling the review of our manuscript. We are sending you our revised manuscript entitled “Interaction of transcription factors BES1/BZR1 in plant growth and stress response”. We appreciate the promoting comments to our paper. We have made modifications to the revised manuscript based on the professional suggestions made by the reviewers. The newly added contents are highlighted in yellow and underlined in the revised manuscript. Here are the point-by-point responses for the reviewers’ comments (reviewers’ comments in black, our replies in blue).

Reviewer 2

The paper is well written and the information is complete and comprehensible. The major problem is that is an only text review, and this decreases the interest. Most readers are looking for figures or tables that facilitate the understanding of the question. In the present version there is only a single table, with a very bad selection of colors, and not really informative. I recommend to complete this review as follows:

  1. a) Include one table summarizing all the published information of BES/BZR1 on plant growth and development, including the effect, the plant where was observed, and the proper citation. 

    Response: Thank you for your professional advice. We have included a table summarizing the interaction proteins of BES1/BZR1 in the regulation of plant growth and development. Please see page 3-4, Table1 and line 91-92.

  1. b) Include one table summarizing all the published information of BES/BZR1 on biotic and abiotic stress, including the effect, the plant where was observed, and the proper citation. 

Response: Thanks for the professional advice. We have included a table summarizing the interaction proteins of BES1/BZR1 involved in the regulation of plant biotic and abiotic stress. Please refer to page 9-10, Table 2 and line 380-384.

  1. c) Include a figure with one plant, indicating in which parts of the plant anatomy the BES/BZR1 have some effect.

Response: Thank you for your professional advice. We have included a figure illustrating the role of BZR1/BES1 in plant growth and development, as well as stress tolerance. Please refer to page 2, Figure 1 and line 63-70.

Minor points: line 328. "S. exigua" should go in italics and with spacing.

Response: Thank you for your professional advice. This mistake has been corrected.

Reviewer 2 Report

Comments and Suggestions for Authors

Manuscript ijms-3049349 "Interplay of transcription factors BES1/BZR1 in plant growth and stress response" by Cao et al. presents an interesting review about plant growth and development, and stress responses in plants.

Overall, writing is clear although with some grammar fails  and the review is sound. Conclusions are mostly supported by the data provided although must be highlighted. However instead of the good review performed, the present version of the manuscript presents some deficiencies which must be revised before publication in IJMS.

English grammar and expression must be revised.

Title should be revised as “Interaction of BES1/BZR1 transcription factors in plant growth and stress response”.

Objectives of the review are not clear. At the end of the Introduction section authors must indicated in a separated paragraph the objectives of the review.

Figure 1 must be incorporated in the main text. However, this information is out of the objective of the review and must be part of the introduction. In addition, this figure must be correctly introduced by authors in this Introduction section.

Around the whole manuscript he names of genes must be in italics.

After the first description of a plant species authors must indicate the botanic origin in italics.

Two new figures (or one) must be incorporated summarizing the main hypothesis about BES1/BZR1 transcription factors and plant development and BES1/BZR1 transcription factors and stress response.

Conclusion section must be revised indicating main implications of the obtained results from a breeding and production point of view.

Comments on the Quality of English Language

English grammar and expression must be revised.

Author Response

Dear Editor and Reviewers,

Thank you very much for handling the review of our manuscript. We are sending you our revised manuscript entitled “Interaction of transcription factors BES1/BZR1 in plant growth and stress response”. We appreciate the promoting comments to our paper. We have made modifications to the revised manuscript based on the professional suggestions made by the reviewers. The newly added contents are highlighted in yellow and underlined in the revised manuscript. Here are the point-by-point responses for the reviewers’ comments (reviewers’ comments in black, our replies in blue).

Reviewer 1

Manuscript ijms-3049349 "Interplay of transcription factors BES1/BZR1 in plant growth and stress response" by Cao et al. presents an interesting review about plant growth and development, and stress responses in plants.

Overall, writing is clear although with some grammar fails and the review is sound. Conclusions are mostly supported by the data provided although must be highlighted. However instead of the good review performed, the present version of the manuscript presents some deficiencies which must be revised before publication in IJMS.

Comment 1. English grammar and expression must be revised.

Response: Thank you for your advice to improve the grammar and expression. The authors have carefully checked all details of the manuscript. We believe that the expressions in this article have been significantly improved to avoid grammar errors or inappropriate expressions.

Comment 2. Title should be revised as “Interaction of BES1/BZR1 transcription factors in plant growth and stress response”.

Response: Thank you for the advice, we have revised the title.

Comment 3. Objectives of the review are not clear. At the end of the Introduction section authors must indicated in a separated paragraph the objectives of the review.

Response: Thanks for your professional advice. The corresponding contents have been added in the revised manuscript. Please see page 2-3, line 71-77.

Comment 4. Figure 1 must be incorporated in the main text. However, this information is out of the objective of the review and must be part of the introduction. In addition, this figure must be correctly introduced by authors in this Introduction section.

    Response: Thanks for your professional advice. We have removed the “Figure 1” in the manuscript and have included two additional tables and a new figure to improve our revised manuscript. The new figure, now labeled as “Figure 1” is displayed on page 2, while the two tables are presented on page 3-4 and page 9-10, respectively.

Comment 5. Around the whole manuscript he names of genes must be in italics.

    Response: Thank you for the reminder. These mistakes have already been corrected.

Comment 6. After the first description of a plant species authors must indicate the botanic origin in italics.

    Response: Thank you for the reminder. These mistakes have already been corrected.

Comment 7.Two new figures (or one) must be incorporated summarizing the main hypothesis about BES1/BZR1 transcription factors and plant development and BES1/BZR1 transcription factors and stress response.

    Response: Thank you for the professional advice. We have included one figure and two tables in our revised manuscript. Please refer to Figure 1 on page 2, Figure legends of Figure 1 on page 2 (line 63-70), Table 1 on page 3-4 and Table 2 on page 9-10.

Comment 8.Conclusion section must be revised indicating main implications of the obtained results from a breeding and production point of view.

Response: Thank you for your professional advice. We have included relevant details in the “conclusion” section. Please see Page 10, Line 404-407.

Round 2

Reviewer 1 Report

Comments and Suggestions for Authors

Authors hace followed my suggestions.

I recommend publication

Reviewer 2 Report

Comments and Suggestions for Authors

Authros ahve revised correctly the manuscript